# DEEP COORDINATION GRAPHS

## ABSTRACT

This paper introduces the *deep coordination graph* (DCG) for collaborative multi-agent reinforcement learning. DCG strikes a flexible trade-off between representational capacity and generalization by factorizing the joint value function of all agents according to a coordination graph into payoffs between pairs of agents. The value can be maximized by local message passing along the graph, which allows training of the value function end-to-end with $Q$-learning. Payoff functions are approximated with deep neural networks and parameter sharing improves generalization over the state-action space. We show that DCG can solve challenging predator-prey tasks that are vulnerable to the *relative overgeneralization* pathology and in which all other known value factorization approaches fail.

## 1 INTRODUCTION

One of the central challenges in cooperative *multi-agent reinforcement learning* (MARL, Oliehoek & Amato, 2016) is coping with the size of the joint action space, which grows exponentially in the number of agents. For example, this paper evaluates tasks where eight agents each have six actions to choose from, yielding a joint action space with more than a million actions. Efficient MARL methods must thus be able to generalize over large joint action spaces, in the same way that convolutional neural networks allow deep RL to generalize over large visual state spaces.

Even though few benchmark tasks actually require agent policies to be independently executable, one common approach to coping with large action spaces is to *decentralize* the decision policy and/or value function. For example, Figure 1a shows how the joint value function can be factorized into *utility functions* that each depend only on the actions of one agent (Sunehag et al., 2018; Rashid et al., 2018). Consequently, the joint value function can be efficiently maximized if each agent simply selects the action that maximizes its corresponding utility function. This factorization can represent any deterministic policy and thus can represent at least one optimal policy. However, that policy may not be learnable due to a game-theoretic pathology called *relative overgeneralization*[1] (Panait et al., 2006): during exploration other agents act randomly and punishment caused by uncooperative agents may outweigh rewards that would be achievable with coordinated actions. If the employed value function does not have the representational capacity to distinguish the values of coordinated and uncoordinated actions, an optimal policy cannot be learned.

However, Castellini et al. (2019) show that higher-order factorization of the value function works surprisingly well in one-shot games that are vulnerable to relative overgeneralization, even if each factor depends on the actions of only a small subset of agents. Such a higher-order factorization can be expressed as an undirected *coordination graph* (CG, Guestrin et al., 2002a), where each vertex represents one agent and each (hyper-)edge one *payoff function* over the joint action space of the connected agents. Figure 1b shows a CG with pairwise edges and the corresponding value factorization. Depending on the CG topology, the value can thus depend nontrivially on the actions of *all* agents, yielding a richer representation. Although the value can no longer be maximized by each agent individually, the greedy action can be found using message passing along the edges (also known as *belief propagation*, Pearl, 1988). *Sparse cooperative Q-learning* (Kok & Vlassis, 2006) applies CGs to MARL but does not scale to modern benchmarks, as each payoff function ($f^{12}$ and $f^{23}$ in Figure 1b) is represented as a table over the state and joint action space of the connected agents. Castellini et al. (2019) use neural networks to approximate payoff functions, but only in one-shot games, and still require a unique function for each edge in the CG. Consequently, each agent group, represented by an edge, must still experience all corresponding action combinations, which can require executing a significant subset of the joint action space.

---

[1] Not to be confused with *generalization* in the context of function approximation.

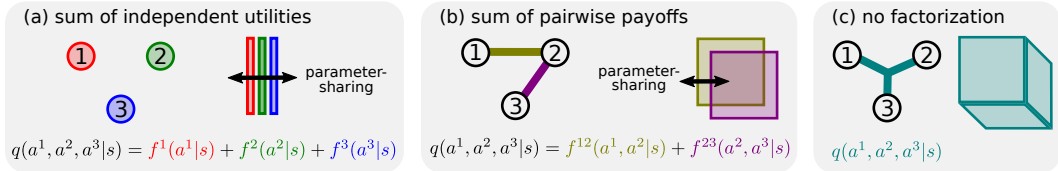

Figure 1: Examples of value factorization for 3 agents: (a) sum of independent utilities (as in VDN, Sunehag et al., 2018) corresponds to an unconnected CG. QMIX uses a monotonic mixture of utilities instead of a sum (Rashid et al., 2018); (b) sum of pairwise payoffs (Castellini et al., 2019), which correspond to pairwise edges; (c) no factorization (as in QTRAN, Son et al., 2019) corresponds to one hyper-edge connecting all agents. Factorization allows parameter sharing between factors, shown next to the CG, which can dramatically improve the algorithm's sample complexity.

To address these issues, this paper proposes the *deep coordination graph* (DCG), a deep RL algorithm that scales to modern benchmark tasks. DCG represents the value function as a CG with pairwise payoffs[2] (Figure 1b) and individual utilities (Figure 1a). This improves the representational capacity beyond state-of-the-art value factorization approaches like VDN (Sunehag et al., 2018) and QMIX (Rashid et al., 2018). To achieve scalability, DCG employs parameter sharing between payoffs and utilities. Parameter sharing has long been a staple of factorized MARL. Methods like VDN and QMIX condition an agent's utility on its history, that is, its past observations and actions, and share the parameters of all utility functions. Experiences of one agent are thus used to train all. This can dramatically improve the sample efficiency compared to unfactored methods (Foerster et al., 2016; 2018; Lowe et al., 2017; Schröder de Witt et al., 2019; Son et al., 2019), which correspond to a CG with one hyper-edge connecting all agents (Figure 1c). DCG takes parameter sharing one step further by approximating *all* payoff functions with the same neural network. To allow unique outputs for each payoff, the network is conditioned on a learned embedding of the participating agents' histories. This requires only one linear layer more than VDN and has thus less parameters as QMIX.

DCG is trained end-to-end with deep $Q$-learning (DQN, Mnih et al., 2015), but uses message passing to coordinate greedy action selection between *all* agents in the graph. For $k$ message passes over $n$ agents with $m$ actions each, the time complexity of maximization is only $\mathcal{O}(km(n+m)|\mathcal{E}|)$, where $|\mathcal{E}| \leq \frac{n^2-n}{2}$ is the number of (pairwise) edges, compared to $\mathcal{O}(m^n)$ for DQN without factorization.

We compare DCG's performance with that of other MARL $Q$-learning algorithms in a challenging family of predator-prey tasks that require coordinated actions. Here DCG is the only algorithm that solves the harder tasks. We also investigate the influence of graph topologies on the performance.

## 2  RELATED WORK

A general overview over cooperative deep MARL can be found in OroojlooyJadid & Hajinezhad (2019). Independent $Q$-learning (IQL Tan, 1993) decentralizes the agents' policy by modeling each agent as an independent $Q$-learner. However, the task from the perspective of a single agent becomes nonstationary as other agents change their policies. To address this, Foerster et al. (2017) show how to stabilize IQL when using experience replay buffers. Another approach to decentralized agents is *centralized training and decentralized execution* (Foerster et al., 2016) with a *factorized value function*. Value decomposition networks (VDN, Sunehag et al., 2018) performs central $Q$-learning with a value function that is the sum of independent utility functions for each agent (Figure 1a). The greedy policy can be executed by maximizing each utility independently. QMIX (Rashid et al., 2018) improves upon this approach by combining the agents' utilities with a mixing network, which is monotonic in the utilities and depends on the global state. This allows different mixtures in different states and the central value can be maximized independently due to monotonicity. All of these approaches are derived in Appendix A.1 and can use parameter sharing between the value/utility functions. However, they represent the joint value with independent values/utilities and are therefore susceptible to the relative overgeneralization pathology. We demonstrate this by comparing DCG with all the above algorithms.

---

[2] The method can be generalized to CG with hyper-edges, that is, to payoff functions for more than 2 agents.

Another straightforward way to decentralize in MARL is to define the joint policy as a product of independent agent policies. This lends itself to the actor-critic framework, where the critic is discarded during execution and can therefore condition on the global state and all agents' actions during training. Examples are MADDPG (Lowe et al., 2017) for continuous actions and COMA (Foerster et al., 2018) for discrete actions. Wei et al. (2018) specifically investigate the relative overgeneralization pathology in continuous multi-agent tasks and show improvement over MADDPG by introducing policy entropy regularization. MACKRL (Schröder de Witt et al., 2019) follows the approach in Foerster et al. (2018), but uses *common knowledge* to coordinate agents during centralized training. Son et al. (2019) define QTRAN, which also has a centralized critic but uses a greedy actor w.r.t. a VDN factorized function. The corresponding utility functions are distilled from the critic under constraints that ensure proper decentralization. Böhmer et al. (2019) present another approach to decentralize a centralized value function, which is locally maximized by coordinate ascent and decentralized by training IQL agents from the same replay buffer. Centralized joint $Q$-value functions do not allow to share parameters to the same extent as value factorization, and we compare DCG to QTRAN to demonstrate the advantage in sample efficiency. That being said, DCG value factorization can in principle be applied to any of the above centralized critics to equally improve sample efficiency at the same cost of representational capacity. We leave this to future work.

Other work deals with gigantic numbers of agents, which requires additional assumptions to reduce the sample complexity. For example, Yang et al. (2018) introduce *mean-field multi-agent learning* (MF-MARL), which factorizes a tabular value function for hundreds of agents into pairwise payoff functions between neighbors in a uniform grid of agents. These payoffs share parameters similar to DCG. Chen et al. (2018) introduce a value factorization for a similar setup based on a low-rank approximation of the joint value. This approach is restricted by uniformity assumptions between agents, but uses otherwise parameter sharing similar to DCG. The value function cannot be maximized globally and must be locally maximized with coordinate ascent. These techniques are designed for much larger sets of agents and do not perform well in the usual MARL settings considered in this paper. While they use similar parameter sharing techniques as DCG, we do therefore not compare against them.

Coordination graphs (CG) have been extensively studied in multi-agent robotics with given payoffs (e.g. Rogers et al., 2011; Yedidsion et al., 2018). Sparse cooperative $Q$-learning (SCQL, Kok & Vlassis, 2006) uses CG in discrete state and action spaces by representing all utility and payoff functions as tables. However, the tabular approach restricts practical application of SCQL to tasks with few agents and small state and action spaces. Castellini et al. (2019) use neural networks to approximate payoff functions, but only in one-shot games, and still require a unique function for each edge in the CG. DCG expands greatly upon these works by introducing parameter sharing between all payoffs (as in VDN/QMIX), conditioning on local information (as in MF-MARL) and evaluating in more complex tasks that are vulnerable to relative overgeneralization.

## 3 BACKGROUND

In this paper we assume a Dec-POMDP for $n$ agents $\langle \mathcal{S}, \{\mathcal{A}^i\}_{i=1}^n, P, r, \{\mathcal{O}^i\}_{i=1}^n, \{\sigma^i\}_{i=1}^n, n, \gamma \rangle$ (Oliehoek & Amato, 2016). $\mathcal{S}$ denotes a finite or continuous set of environmental states and $\mathcal{A}^i$ the discrete set of actions available to agent $i$. At discrete time $t$, the next state $s_{t+1} \in \mathcal{S}$ is drawn from transition kernel $s_{t+1} \sim P(\cdot|s_t, \boldsymbol{a}_t)$, conditioned on the current state $s_t \in \mathcal{S}$ and joint action $\boldsymbol{a}_t \in \mathcal{A} := \mathcal{A}^1 \times \ldots \times \mathcal{A}^n$ of all agents. A transition yields collaborative reward $r_t := r(s_t, \boldsymbol{a}_t)$, and $\gamma \in [0, 1)$ denotes the discount factor. Each agent $i$ observes the state only partially by drawing observations $o_t^i \in \mathcal{O}^i$ from its observation kernel $o_t^i \sim \sigma^i(\cdot|s_t)$. The history of agent $i$'s observations $o_t^i \in \mathcal{O}^i$ and actions $a_t^i \in \mathcal{A}^i$ is in the following denoted as $\tau_t^i := (o_0^i, a_0^i, o_1^i, \ldots, o_{t-1}^i, a_{t-1}^i, o_t^i) \in (\mathcal{O}^i \times \mathcal{A}^i)^t \times \mathcal{O}^i$. Without loss of generality, this paper restricts itself to episodic tasks, which yield episodes $(s_0, \{o_0^i\}_{i=1}^n, \boldsymbol{a}_0, r_0, \ldots, s_T, \{o_T^i\}_{i=1}^n)$ of varying (but finite) length $T$.

### 3.1 DEEP $Q$-LEARNING

The goal of collaborative multi-agent reinforcement learning (MARL) is to find an optimal policy $\pi^* : \mathcal{S} \times \mathcal{A} \to [0, 1]$, that chooses joint actions $\boldsymbol{a}_t \in \mathcal{A}$ such that the expected discounted sum of future reward is maximized. This can be achieved by estimating the optimal $Q$-value function[3]:

---

[3] We overload the notation $f(y|x)$ to also indicate the inputs $x$ and outputs $y$ of multivariate functions $f$.

$$q^*(\boldsymbol{a}|s) \quad := \quad \mathbb{E}_{\pi^*}\Big[ \sum_{t=0}^{T-1} \gamma^t r_t \,\Big|\, {}^{s_0=s}_{\boldsymbol{a}_0=\boldsymbol{a}} \Big] \quad = \quad r(s,\boldsymbol{a}) + \gamma \int P(s'|s,\boldsymbol{a}) \max q^*(\cdot|s')\, ds' \,. \quad (1)$$

The optimal policy $\pi^*(\cdot|s_t)$ chooses greedily the action $\boldsymbol{a} \in \mathcal{A}$ that maximizes the corresponding optimal $Q$-value $q^*(\boldsymbol{a}|s_t)$. In fully observable discrete state and action spaces, $q^*$ can be learned in the limit from interactions with the environment (Watkins & Dayan, 1992). For large or continuous state spaces, $q^*$ can only be approximated, e.g., with a deep neural network $q_\theta$ (DQN, Mnih et al., 2015), parameterized by $\theta$, by minimizing the mean-squared Bellman loss with gradient descent:

$$\mathcal{L}_{\text{DQN}} \quad := \quad \mathbb{E}\Big[ \tfrac{1}{T} \sum_{t=0}^{T-1} \Big( r_t + \gamma \max q_{\bar{\theta}}(\cdot|s_{t+1}) - q_\theta(\boldsymbol{a}_t|s_t) \Big)^2 \,\Big|\, \{s_t, \boldsymbol{a}_t, r_t, s_{t+1}\}_{t=0}^{T} \sim D \Big]. \quad (2)$$

The expectation is estimated with samples from an experience replay buffer $D$ holding previously observed episodes (Lin, 1992), and $\bar{\theta}$ denotes the parameter of a separate target network, which is periodically replaced with a copy of $\theta$ to improve stability. Double $Q$-learning further stabilizes training by choosing the next action greedily w.r.t. the current network $q_\theta$, i.e., $q_{\bar{\theta}}(\arg\max q_\theta(\cdot|s_{t+1})|s_{t+1})$ instead of the target network $\max q_{\bar{\theta}}(\cdot|s_{t+1})$ (van Hasselt et al., 2016).

In partially observable environments, the learned policy cannot condition on the state $s_t$. Instead, Hausknecht & Stone (2015) approximate a $Q$-function that conditions on the agent's history $\boldsymbol{\tau}_t := \{\tau_t^i\}_{i=1}^n$, i.e., $q_\theta(\boldsymbol{a}|\boldsymbol{\tau}_t)$, by conditioning a recurrent neural network (e.g., a GRU, Chung et al., 2014) on the agents' observations $\boldsymbol{o}_t := (o_t^1, \ldots, o_t^n)$ and last actions $\boldsymbol{a}_{t-1}$, that is, $q_\theta(\boldsymbol{a}|\boldsymbol{h}_t)$ conditions on the recurrent network's hidden state $h_\psi(\boldsymbol{h}_t|\boldsymbol{h}_{t-1}, \boldsymbol{o}_t, \boldsymbol{a}_{t-1})$, where $\boldsymbol{h}_0$ is initialized with zeros.

Applying DQN to multi-agent tasks quickly becomes infeasible, due to the combinatorial explosion of state and action spaces. Moreover, DQN value functions cannot be maximized without evaluating all possible actions. To allow MARL $Q$-learning with efficient maximization, various algorithms based on value factorization have been developed. We derive IQL (Tan, 1993), VDN (Sunehag et al., 2018), QMIX (Rashid et al., 2018) and QTRAN (Son et al., 2019) in Appendix A.1.

### 3.2 COORDINATION GRAPHS

An undirected *coordination graph* (CG, Guestrin et al., 2002a) $\mathcal{G} = \langle \mathcal{V}, \mathcal{E} \rangle$ contains a vertex $v_i \in \mathcal{V}$ for each agent $1 \le i \le n$ and a set of undirected edges $\{i,j\} \in \mathcal{E}$ between vertices $v_i$ and $v_j$. The graph is usually specified before training, but Guestrin et al. (2002b) suggest that the graph could also depend on the state, that is, each state can have its own unique CG. A CG induces a factorization[4] of the $Q$-function into *utility functions* $f^i$ and *payoff functions* $f^{ij}$ (Fig. 1a and 1b):

$$q^{\text{CG}}(s_t, \boldsymbol{a}) \quad := \quad \frac{1}{|\mathcal{V}|} \sum_{v^i \in \mathcal{V}} f^i(a^i|s_t) \,+\, \frac{1}{|\mathcal{E}|} \sum_{\{i,j\} \in \mathcal{E}} f^{ij}(a^i, a^j|s_t) \,. \quad (3)$$

The special case $\mathcal{E} = \varnothing$ yields VDN, but each additional edge enables the representation of the value of the actions of a pair of agents and can thus help to avoid relative overgeneralization. Prior work also considered higher order coordination where the payoff functions depend on arbitrary sets of actions (Guestrin et al., 2002a; Kok & Vlassis, 2006; Castellini et al., 2019), corresponding to graphs with hyper-edges (Figure 1c). For the sake of simplicity we restrict ourselves here to pairwise edges, which yield at most $|\mathcal{E}| \le \frac{1}{2}(n^2 - n)$ edges, in comparison to up to $\frac{n!}{d!\,(n-d)!}$ hyper-edges of degree $d$. The induced $Q$-function $q^{\text{CG}}$ can be maximized locally using *max-plus*, also known as *belief propagation* (Pearl, 1988). At time $t$ each node sends messages $\mu_t^{ij}(a^j) \in \mathbb{R}$ over all adjacent edges $\{i,j\} \in \mathcal{E}$, which can be computed locally:

$$\mu_t^{ij}(a^j) \quad \leftarrow \quad \max_{a^i} \Big\{ \tfrac{1}{|\mathcal{V}|} f^i(a^i|s_t) + \tfrac{1}{|\mathcal{E}|} f^{ij}(a^i, a^j|s_t) + \sum_{\{k,i\} \in \mathcal{E}} \mu_t^{ki}(a^i) - \mu_t^{ji}(a^i) \Big\}. \quad (4)$$

This process repeats for a number of iterations, after which each agent $i$ can locally find the action $a_*^i$ that maximizes the estimated $Q$-value:

$$a_*^i \quad := \quad \arg\max_{a^i} \Big\{ \tfrac{1}{|\mathcal{V}|} f^i(a^i|s_t) + \sum_{\{k,i\} \in \mathcal{E}} \mu_t^{ki}(a^i) \Big\}. \quad (5)$$

---

[4] The normalizations $\frac{1}{|\mathcal{V}|}$ and $\frac{1}{|\mathcal{E}|}$ are not strictly necessary, but allow to potentially generalize to other CGs.

Convergence of messages is guaranteed for acyclic CGs (Pearl, 1988; Wainwright et al., 2004), but messages can diverge in cyclic graphs, for example fully connected CGs. Subtracting a normalization constant $c_{ij} := \sum_a \mu_t^{ij}(a) / |\mathcal{A}^i|$ from each message $\mu^{ij}$ before it is sent often leads to convergence in practice (Murphy et al., 1999; Crick & Pfeffer, 2002; Yedidia et al., 2003). See Algorithm 3 in the appendix for details.

# 4 METHOD

We now introduce the *deep coordination graph* (DCG), which learns the utility and payoff functions of a coordination graph $\langle \mathcal{V}, \mathcal{E} \rangle$ with deep neural networks. A direct implementation as in Castellini et al. (2019) would learn a separate network for each function $f^i$ and $f^{ij}$. However, properly approximating these $Q$-values requires observing the joint actions of each agent pair in the edge set $\mathcal{E}$, which for dense graphs can be a significant subset of the joint action space of all agents $\mathcal{A}$. We address this issue by focusing on an architecture that shares parameters across functions and restricts them to locally available information, i.e., to the histories of the participating agents.

Sunehag et al. (2018) introduces parameter sharing between the agents' utility functions $f^i(u^i|s_t) \approx f_\theta^v(u^i|\tau_t^i)$ to dramatically improve the sample efficiency of VDN. Agents can have different action spaces $\mathcal{A}^i$ but the choice of unavailable actions during maximization can be prevented by setting the utilities of unavailable actions to $-\infty$. Specialized roles for individual agents can be achieved by conditioning $f_\theta^v$ on the agent's role, or more generally on the agent's ID (Foerster et al., 2018; Rashid et al., 2018). The DCG uses the same utility functions and adds payoff functions specified by pairwise edges in a given CG (Guestrin et al., 2002a). We take inspiration from highly scalable methods (Yang et al., 2018; Chen et al., 2018) and improve over SCQL (Kok & Vlassis, 2006) and the approach of Castellini et al. (2019) by incorporating the following design principles:

   i. restricting the payoffs $f^{ij}(a^i, a^j|\tau_t^i, \tau_t^j)$ to local information of agents $i$ and $j$ only;
   ii. sharing parameters between all payoff and utility functions through a common RNN;
   iii. low-rank approximation of payoff matrices $f^{ij}(\cdot, \cdot|\tau_t^i, \tau_t^j)$ for large action spaces;
   iv. allowing transfer/generalization to different CG (as suggested in Guestrin et al., 2002b);
   v. allowing the use of privileged information like the global state during training.

Restricting the payoff's input (i) and sharing parameters (ii) improves sample efficiency significantly. As in Sunehag et al. (2018), all utilities are computed with the same neural network $f_\theta^i(u^i|\tau_t^i) \approx f_\theta^v(u^i|h_t^i)$, but unlike Castellini et al. (2019), all payoffs are computed with the same neural network $f^{ij}(a^i, a^j|\tau_t^i, \tau_t^j) \approx f_\phi^e(a^i, a^j|h_t^i, h_t^j)$, too. Both share parameters though a common RNN $h_t^i := h_\psi(\cdot|h_{t-1}^i, o_t^i, a_{t-1}^i)$, which is initialized with $h_0^i := h_\psi(\cdot|\mathbf{0}, o_0^i, \mathbf{0})$.

Modeling the payoff function $f_\phi^e$ similar to DQN (Mnih et al., 2015) yields $|\mathcal{A}^i \times \mathcal{A}^j|$ separate outputs for edge $\{i, j\}$. For example, each agent in a StarCraft 2 map with 8 enemies has 13 actions (SMAC, Samvelyan et al., 2019), which yields 169 outputs of $f_\phi^e$. As only executed actions-pairs are updated during $Q$-learning, the parameters of many outputs remain unchanged for long stretches of time, while the underlying RNN $h_\psi$ keeps evolving. This can slow down training and affect message passing. To reduce the number of parameters and improve the frequency in which they are updated, we propose a *low-rank approximation* of the payoff (iv) with rank $K$, similar to Chen et al. (2018):

$$ f_\phi^e(a^i, a^j|h_t^i, h_t^j) \quad := \quad \sum_{k=1}^K \hat{f}_{\hat{\phi}}^e(a^i, k|h_t^i, h_t^j) \, \bar{f}_{\bar{\phi}}^e(a^j, k|h_t^i, h_t^j) . \tag{6} $$

The approximation can be computed in one forward-pass with $K(|\mathcal{A}^i| + |\mathcal{A}^j|)$ outputs and parameters $\phi := \{\hat{\phi}, \bar{\phi}\}$. Note that a rank $K = \min\{|\mathcal{A}^i|, |\mathcal{A}^j|\}$ approximation does not restrict the output's expressiveness, while lower ranks share parameters and updates to speed up learning.

Generalization (or zero-shot transfer) of the learned functions onto new CGs in (iv) poses some practical design challenges. To be applicable to different graphs/topologies, DCG must be invariant to reshuffling of agent indices. This requires the payoff matrix $\boldsymbol{f}^{ij}$, of dimensionality $|\mathcal{A}^i| \times |\mathcal{A}^j|$, to be the same as $(\boldsymbol{f}^{ji})^\top$ with swapped inputs. We enforce invariance by computing the function $f_\phi^e$ for both combinations and use the average between the two. Note that this retains the ability to learn asymmetric payoff matrices $\boldsymbol{f}^{ij} \neq (\boldsymbol{f}^{ij})^\top$. However, this paper does not evaluate (iv) and we leave the transfer of a learned DCG onto different graphs to future work. The DCG $Q$-value function is:

$$q_{\theta\phi\psi}^{\text{DCG}}(\boldsymbol{\tau}_t, \boldsymbol{a}) \; := \; \frac{1}{|\mathcal{V}|}\sum_{i=1}^{n}\underbrace{f_\theta^v(a^i|\boldsymbol{h}_t^i)}_{f_{i,a^i}^{\text{V}}} + \frac{1}{2|\mathcal{E}|}\sum_{\{i,j\}\in\mathcal{E}}\Big(\underbrace{f_\phi^e(a^i,a^j|\boldsymbol{h}_t^i,\boldsymbol{h}_t^j) + f_\phi^e(a^j,a^i|\boldsymbol{h}_t^j,\boldsymbol{h}_t^i)}_{f_{\{i,j\},a^i,a^j}^{\text{E}}}\Big). \quad (7)$$

Moreover, some tasks allow access to privileged information like the global state $s_t \in \mathcal{S}$ during training (but not execution). We therefore propose in (v) to use this information in a *privileged bias function* $v_\varphi : \mathcal{S} \to \mathbb{R}$ with parameters $\varphi$, that is, $q_{\theta\phi\psi\varphi}^{\text{DCG-V}}(s_t, \boldsymbol{\tau}_t, \boldsymbol{a}) := q_{\theta\phi\psi}^{\text{DCG}}(\boldsymbol{\tau}_t, \boldsymbol{a}) + v_\varphi(s_t)$.

We train DCG end-to-end with the DQN loss in (2) and Double $Q$-learning (van Hasselt et al., 2016). Given the tensors $\boldsymbol{f}^{\text{V}} \in \mathbb{R}^{|\mathcal{V}|\times A}$ and $\boldsymbol{f}^{\text{E}} \in \mathbb{R}^{|\mathcal{E}|\times A\times A}$, $A := |\cup_i \mathcal{A}^i|$, where all unavailable actions are set to $-\infty$, the $Q$-value can be maximized by message passing as defined in (4) and (5). The detailed procedure of computing the tensors (Algorithm 1), the $Q$-value (Algorithm 2) and greedy action selection (Algorithm 3) is given in the appendix. Note that no gradients flow through the message passing loop, as DQN maximizes only the bootstrapped future value.

The key benefit of DCG lies in its ability to prevent *relative overgeneralization* during the exploration of agents: take the example of two hunters who have cornered their prey. The prey is dangerous and attempting to catch it alone can lead to serious injuries. From the perspective of each hunter, the expected reward for an attack depends on the actions of the other agent, who will initially behave randomly. If the punishment for attacking alone outweighs the reward for catching the prey, agents that cannot represent the value for joint actions (QMIX, VDN, IQL) cannot learn the optimal policy. However, estimating a value function over the joint action space (as in QTRAN) can be equally prohibitive, as it requires many more samples for the same prediction quality. DCG provides a flexible function class between these extremes that can be tailored to the task at hand.

## 5 VALIDATION

In this section we compare the performance of DCG with various topologies (see Table 1) to the state-of-the-art algorithms QTRAN (Son et al., 2019), QMIX (Rashid et al., 2018), VDN (Sunehag et al., 2018) and IQL (Tan, 1993). We also evaluate a CG baseline similar to Castellini et al. (2019), which conditions on a shared RNN which summarizes all agents' histories. All algorithms are implemented in the multi-agent framework PYMARL (Samvelyan et al., 2019).

| | |
|---|---|
| DCG | $\mathcal{E} := \big\{\{i,j\}\,\big|\,1 \le i < n, i < j \le n\big\}$ |
| CYCLE | $\mathcal{E} := \big\{\{i,(i \bmod n)+1\}\,\big|\,1 \le i \le n\big\}$ |
| LINE | $\mathcal{E} := \big\{\{i,i+1\}\,\big|\,1 \le i < n\big\}$ |
| STAR | $\mathcal{E} := \big\{\{1,i\}\,\big|\,2 \le i \le n\big\}$ |
| VDN | $\mathcal{E} := \varnothing$ |

Table 1: Tested graph topologies for DCG.

We evaluate these methods in two complex grid-world tasks: the first formulates the relative overgeneralization problem as a family of predator-prey tasks and the second investigates how *artificial decentralization* can hurt tasks that demand non-local coordination between agents. In the latter case, decentralized value functions (QMIX, VDN, IQL) cannot learn coordinated action selection between agents that cannot see each other directly and thus converge to a sub-optimal policy. We also evaluate DCG and DCG-V in StarCraft 2 micromanagement tasks from the StarCraft Multi-Agent Challenge (SMAC, Samvelyan et al., 2019).

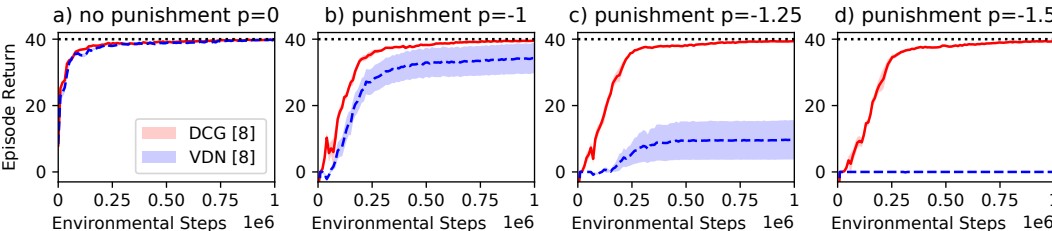

Figure 2: Influence of punishment $p$ for attempts to catch prey alone on greedy test episode return (mean and shaded standard error, [number of seeds]) in a coordination task where 8 agents hunt 8 prey (dotted line denotes best possible return). Note that fully connected DCG (DCG, solid) are able to represent the value of joint actions and coordinate maximization, which leads to a better performance for larger $p$, where DCG without edges (VDN, dashed) has to fail eventually ($p < -1$).

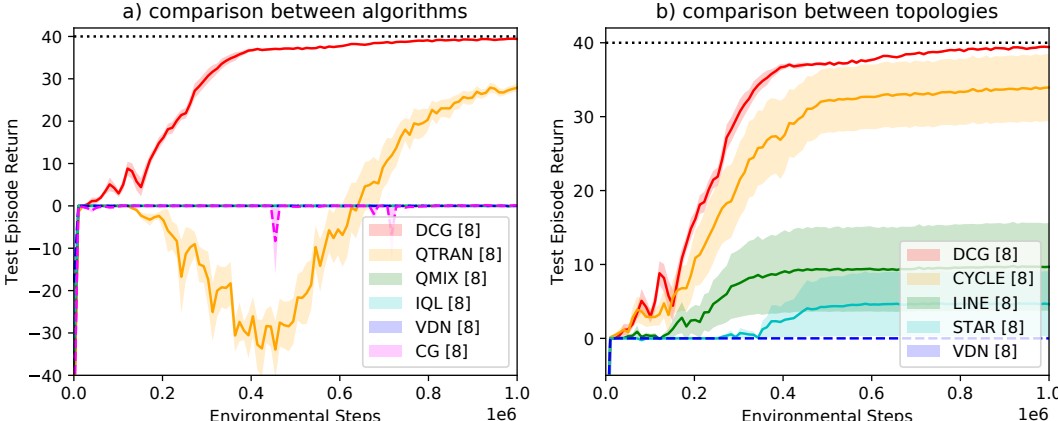

Figure 3: Greedy test episode return for the coordination task of Figure 2 with punishment $p = -2$: (a) comparison to baseline algorithms; (b) comparison between DCG topologies. Note that QMIX, IQL, VDN and CG (dashed) do not solve the task (return 0) due to *relative overgeneralization* and that QTRAN learns very slowly due to the large action space. The reliability of DCG depends on the CG-topology: all seeds with fully connected DCG solved the task, but the high standard error for CYCLE, LINE and STAR topologies is caused by some seeds succeeding while others fail completely.

## 5.1 RELATIVE OVERGENERALIZATION

To model the challenge of relative overgeneralization, we consider a partially observable grid-world predator-prey task: 8 agents have to hunt 8 prey in a $10 \times 10$ grid. Each agent can either move in one of the 4 compass directions, remain still, or try to catch any adjacent prey. Impossible actions, that is, moves into an occupied target position or catching when there is no adjacent prey, are treated as unavailable. The prey moves by randomly selecting one available movement or remains motionless if all surrounding positions are occupied. If two adjacent agents execute the catch action, a prey is caught and both the prey and the catching agents are removed from the grid. An agent's observation is a $5 \times 5$ sub-grid centered around it, with one channel showing agents and another indicating prey. Removed agents and prey are no longer visible and removed agents receive a special observation of all zeros. An episode ends if all agents have been removed or after 200 time steps. Capturing a prey is rewarded $r = 10$, but unsuccessful attempts by single agents are punished by a negative reward $p$. The task is similar to one proposed by Son et al. (2019), but significantly more complex, both in terms of the optimal policy and in the number of agents.

To demonstrate the effect of relative overgeneralization, Figure 2 shows the average return of greedy test episodes for varying punishment $p$ as mean and standard error over 8 independent runs. Without punishment ($p = 0$ in Figure 2a), fully connected DCG (DCG, solid) performs as well as DCG without edges (VDN, dashed). However, for stronger punishment VDN becomes more and more unreliable, which is visible in the large standard errors in Figures 2b and 2c, until it fails completely for $p \leq -1.5$ in Figure 2d. This is due to relative overgeneralization, as VDN cannot represent the values of joint actions during exploration. DCG, on the other hand, learns only slightly slower with punishment and converges otherwise reliably to the optimal solution (dotted line).

Figure 3a shows how well DCG performs in comparison with the baseline algorithms in Appendix A.1 for a strong punishment of $p = -2$. Note that QMIX, IQL and VDN completely fail to learn the task (return 0) due to their restrictive value factorization. While CG could in principle learn the same policy as DCG, the lack of parameter sharing appears to slow down learning dramatically here, which yields no reward in the first million transitions. QTRAN estimates the values with a centralized function, which conditions on all agents' actions, and can therefore learn the task. However, QTRAN requires much more samples than DCG before a useful policy can be learned, due to the size of the joint action space. This is in line with the findings of Son et al. (2019), which required much more samples to learn a task with four agents than with two and also show the characteristic dip in performance with more agents. In comparison with both QTRAN and CG, fully connected DCG (DCG) learn near-optimal policies remarkably fast and reliable.

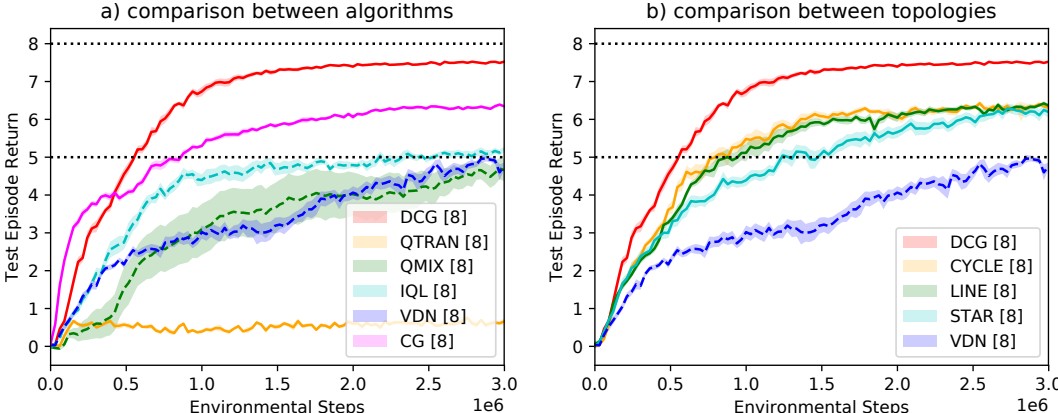

Figure 4: Greedy test episode return (mean and shaded standard error, [number of seeds]) in a *non-decentralizable task* where 8 agents hunt 8 prey: (a) comparison to baseline algorithms; (b) comparison between DCG topologies. The prey turns randomly into punishing ghosts, which are indistinguishable from normal prey. The prey status is only visible at an indicator that is placed randomly at each episode in one of the grid's corners. QTRAN, QMIX, IQL and VDN learn decentralized policies, which are at best suboptimal in this task (around lower dotted line). Fully connected DCG and CG can learn a near-optimal policy (upper dotted line denotes best possible return), but a lack of parameter sharing slows down CG and yields sub-optimal performance in comparison to DCG.

We also investigated the performance of various DCG topologies defined in Table 1. Figure 3b shows that in particular the *reliability* of the achieved test episode return depends strongly on the graph topology. While all seeds of fully connected DCG succeed (DCG), DCG with CYCLE, LINE and STAR topologies have varying means while exhibiting large standard errors. The high deviations are caused by some runs finding near-optimal policies, while others fail completely (return 0). One possible explanation is that for the failed seeds the rewarded experiences, observed in the initial exploration, are only amongst agents that do not share a payoff function. Due to the relative over-generalization pathology, the learned greedy policy no longer explores 'catch' actions and existing payoff functions cannot experience the reward for coordinated actions anymore. It is therefore not surprising that fully connected graphs perform best, as they represent the largest function class and require the fewest assumptions. The topology had also little influence on the runtime of DCG, due to efficient batching on the GPU. The tested fully connected DCG only considers pairwise edges. Hyper-edges between more than two agents (Figure 1c) would yield even richer value representations, but would also require more samples to sufficiently approximate the payoff functions. This effect can be seen in the much slower learning QTRAN results in Figure 3a.

## 5.2 ARTIFICIAL DECENTRALIZATION

The choice of decentralized value functions is in most cased motivated by the huge joint action spaces and not because the task actually requires decentralized execution: it is an artificial decentralization. While this often works surprisingly well, we want to investigate how existing algorithms deal with tasks that cannot be fully decentralized. One obvious case in which decentralization must fail is when the optimal policy cannot be represented by utility functions alone. For example, decentralized policies behave suboptimally in tasks where the optimal policy must condition on multiple agents' observations in order to achieve the best return. Payoff functions in DCG, on the other hand, condition on pairs of agents and can thus represent a richer class of policies. Note that dependencies on more agents can be modeled as hyper-edges in the DCG (Figure 1c), but this hurts the sample efficiency as discussed above.

We evaluate the advantage of a richer policy class with a variation of the above predator-prey task. Inspired by the video game PACMAN, at each turn a fair coin flip decides randomly whether all prey are turned into dangerous *ghosts*. To disentangle the effects of relative overgeneralization, prey can be caught by only one agent (without punishment), yielding a reward of $r = 1$. However, if the agent captures a ghost, the team is punished with $r = -1$. Ghosts are indistinguishable from normal prey, except for a special indicator that is placed in a random corner at the beginning of each episode. The indicator signals on an additional channel of the agents' observations whether the prey are currently

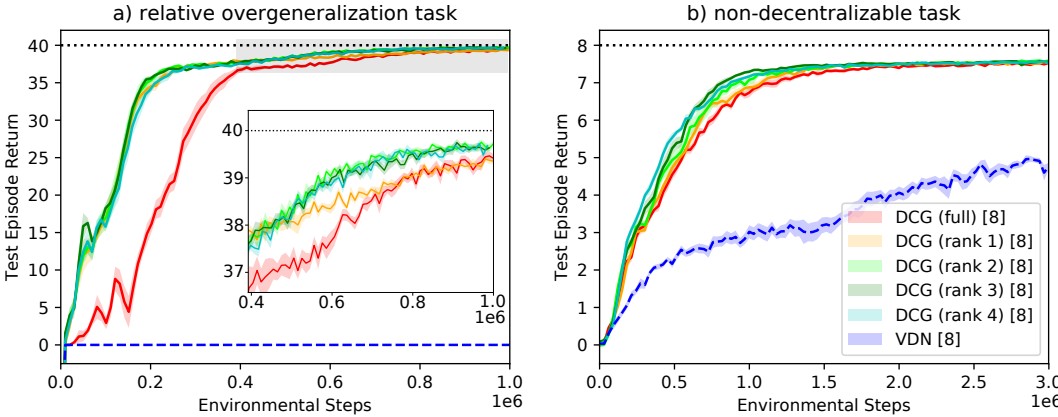

Figure 5: Low-rank payoff approximation for the tasks in (a) Figure 3 and (b) Figure 4. The inlay in (a) magnifies the area of the gray box. Note that in (a) all approximation `ranks` learn much faster than `full` DCG, and `rank 2-4` also have better performance.

ghosts. Due to the short visibility range of the agents, the indicator is only visible in one of the 9 positions closest to its corner.

Figure 4a shows the performance of `QTRAN`, `QMIX`, `IQL` and `VDN`, all of which have decentralized policies, in comparison to fully connected `DCG` and `CG`. The baseline algorithms have to learn a policy that first identifies the location of the indicator and then herds prey into that corner, where the agent is finally able to catch it without risk. By contrast, DCG and CG can learn a policy where one agent finds the indicator, allowing all agents that share an edge to condition their payoffs on that agent's current observation. As a result, this policy can catch prey much more reliably, as seen in the high performance of `DCG` compared to all baseline algorithms. Interestingly, as `CG` conditions on all agents' histories, the baseline shows an advantage in the beginning but then learns much slower and reaches a significantly lower performance. We also investigate the influence of the DCG topologies of Table 1, shown in Figure 4b. Note that while other topologies do not reach the same performance as fully connected DCG, they still reach a policy that significantly outperforms all baseline algorithms, around the same performance as fully connected `CG`.

## 5.3 LOW-RANK APPROXIMATION

While the above experiments already show a significant advantage of DCG with independent payoff outputs for each action pair, we observed some serious performance issues on StarCraft 2 maps with this architecture. The most likely cause is the difference in the number of actions per agent: predator-prey agents choose between $|\mathcal{A}^i| = 6$ actions, whereas SMAC agents on comparable maps with 8 enemies have $|\mathcal{A}^i| = 13$ actions. While payoff matrices with 36 outputs in predator-prey appear reasonable to learn, 169 outputs in StarCraft 2 would require significantly more samples to estimate the payoff of each joint-action properly.

Figure 5 shows the influence of *low-rank payoff approximation* (Equation 6 with $K \in \{1, \ldots, 4\}$) on both predator-prey tasks from previous subsections. One can see in Figure 5a that any low-rank approximation (`DCG (rank K)`) significantly improves the sample efficiency over the default architecture with independent payoff for each action pair (`DCG (full)`). Only rank $K = 1$ leads to slightly lower performance, which can be seen in the inlay plot. We conclude that rank $K \geq 2$ is needed to represent the true values, but rank $K = 1$ already suffices to overcome the relative overgeneralization pathology. The improvement in Figure 5b is less impressive, but shows that even rank $K = 1$ approximation (`DCG (rank 1)`) yields slight performance gains over `DCG (full)`.

## 5.4 STARCRAFT 2

The default architecture of DCG with independent payoff for each action pair performed poorly in StarCraft 2. We therefore tested $K = 1$ low-rank payoff approximation DCG with (`DCG-V`) and without (`DCG`) privileged information bias function $v_\varphi$, as described in Section 4, on six StarCraft 2 maps (from SMAC, Samvelyan et al., 2019). The learning curves for all StarCraft 2 maps are

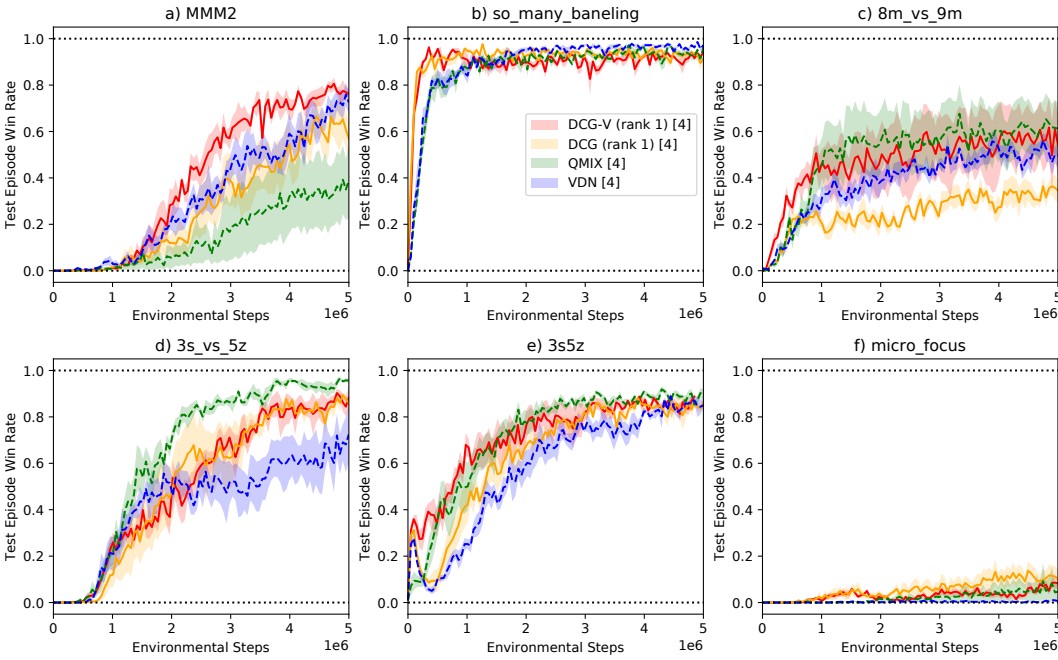

Figure 6: Cumulative reward for test episodes on SMAC maps (mean and shaded standard error, [number of seeds]) for `QMIX`, `VDN` and fully connected DCG with rank $K = 1$ payoff approximation (`DCG (rank 1)`) and additional state-dependent bias function (`DCG-V (rank 1)`).

given in Figure 6. We expected DCG to only yield an advantage on maps which struggle with the relative overgeneralization pathology or some related issue. In this light it is somewhat surprising that both `DCG` and `DCG-V` outperform `VDN` on almost all maps. `DCG-V` performs at least as good as `DCG` and clearly outperforms it on some maps, which demonstrates that privileged information is clearly useful. As expected, a direct comparison with the state-of-the-art method `QMIX` depends strongly on the StarCraft 2 map. On the one hand, `DCG-V` clearly outperforms `QMIX` on `MMM2` (Figure 6a), which is classified as *super hard* by SMAC. We also learn much faster on the *easy* map `so_many_baneling` (Figure 6b). On the other hand, `QMIX` performs better on the *hard* map `3s_vs_5z` (Figure 6d), which might be due to the low number of 3 agents. For this few agents, the added representational capacity of DCG may not improve the task as much as the non-linear state-dependent mixing of QMIX. However, it is hard to pin-point why state dependent mixing is an advantage here.

We conclude from these results that that some maps (like `MMM2`) clearly benefit from the improved coordination and value representation of DCG, while in most others `DCG-V` is on par with `QMIX`.

## 6   CONCLUSIONS & FUTURE WORK

This paper introduced the *deep coordination graph* (DCG), an architecture for value factorization that is specified by a *coordination graph* (CG) and can be maximized by message passing. We evaluated deep $Q$-learning with DCG and show that the architecture enables learning of tasks where *relative overgeneralization* causes all decentralized baselines to fail, whereas centralized critics are much less sample efficient than DCG. We also demonstrated that artificial decentralization can lead to suboptimal behavior in all compared methods except DCG. Our method significantly improves over existing CG methods, which we demonstrate experimentally as well. Fully connected DCG performed best in all experiments and should be preferred in the absence of prior knowledge about the task. Additionally, we introduced a low-rank payoff approximation for large action spaces and a privileged bias function (DCG-V). Evaluated on StarCraft 2 micromanagement tasks, DCG-V performs competitive with the state-of-the-art QMIX. Although not evaluated in this paper, DCG should be able to transfer/generalize to different graphs/topologies and can also be defined for higher-order dependencies. This would in principle allow the training of DCG on dynamically generated graphs, including hyper-edges with varying degrees. We plan to investigate this in future work.

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

## A  APPENDIX

### A.1  BASELINE ALGORITHMS

**IQL**  *Independent Q-learning* (Tan, 1993) is a straightforward approach of value decentralization that allows efficient maximization by modeling each agent as an independent DQN $q_\theta^i(a^i|\tau_t^i)$. The value functions can be trained without any knowledge of other agents, which are assumed to be part of the environment. This violates the stationarity assumption of $P$ and can become therefore instable (see e.g. Foerster et al., 2017). IQL is nonetheless widely used in practice, as parameter sharing between agents can make it very sample efficient.

Note that parameter sharing requires access to privileged information during training, called *centralized training and decentralized execution* (Foerster et al., 2016). This is particularly useful for actor-critic methods like MADDPG (Lowe et al., 2017), Multi-agent soft Q-learning (Wei et al., 2018), COMA (Foerster et al., 2018) and MACKRL (Schröder de Witt et al., 2019), where the centralized critic can condition on the underlying state $s_t$ and the joint action $\boldsymbol{a}_t \in \mathcal{A}$.

**VDN**  Another way to exploit centralized training is *value function factorization*. For example, value decomposition networks (VDN, Sunehag et al., 2018) perform centralized deep $Q$-learning on a joint $Q$-value function that factors as the sum of independent *utility functions* $f^i$, for each agent $i$:

$$q_\theta^{\text{VDN}}(\boldsymbol{\tau}_t, \boldsymbol{a}) \quad := \quad \sum_{i=1}^n f_\theta^i(a^i|\tau_t^i) \,. \tag{8}$$

This value function $q^{\text{VDN}}$ can be maximized by maximizing each agent's utility $f_\theta^i$ independently.

**QMIX**  (Rashid et al., 2018) improves upon this concept by factoring the value function as

$$q_{\theta\phi}^{\text{QMIX}}(s_t, \boldsymbol{\tau}_t, \boldsymbol{a}) \quad := \quad \varphi_\phi\big(s_t, f_\theta^1(a^1|\tau_t^1), \dots, f_\theta^n(a^n|\tau_t^n)\big) \,. \tag{9}$$

Here $\varphi_\phi$ is a *monotonic mixing hypernetwork* with non-negative weights that retains monotonicity in the inputs $f_\theta^i$. Maximizing each utility $f_\theta^i$ therefore also maximizes the joint value $q^{\text{QMIX}}$, as in VDN. The mixing parameters are generated by a neural network, parameterized by $\phi$, that condition on the state $s_t$, allowing different mixing of utilities in different states. QMIX improves performance over VDN, in particular in StarCraft II micromanagement tasks (SMAC, Samvelyan et al., 2019).

**QTRAN**  Recently Son et al. (2019) introduced QTRAN, which learns the centralized critic of a greedy policy w.r.t. a VDN factorized function, which in turn is distilled from the critic by regression under constraints. The algorithm defines three value functions $q^{\text{VDN}}$, $q$ and $v$, where $q(\boldsymbol{\tau}_t, \boldsymbol{u})$ is the centralized Q-value function, as in Section 3.1, and

$$v(\boldsymbol{\tau}_t) \quad := \quad \max q(\boldsymbol{\tau}_t, \cdot) - \max q^{\text{VDN}}(\boldsymbol{\tau}_t, \cdot) \,. \tag{10}$$

They prove that the greedy policies w.r.t. $q$ and $q^{\text{VDN}}$ are identical under the constraints:

$$q^{\text{VDN}}(\boldsymbol{\tau}_t, \boldsymbol{u}) - q(\boldsymbol{\tau}_t, \boldsymbol{a}) + v(\boldsymbol{\tau}_t) \quad \geq \quad 0 \,, \quad \forall \boldsymbol{a} \in \mathcal{A}, \quad \forall \boldsymbol{\tau}_t \in \{(\mathcal{O}^i \times \mathcal{A}^i)^t \times \mathcal{O}^i\}_{i=1}^n, \tag{11}$$

with strict equality if and only if $\boldsymbol{a} = \arg\max q^{\text{VDN}}(\boldsymbol{\tau}_t, \cdot)$. QTRAN minimizes the parameters $\phi$ of the centralized asymmetric value $q_\phi^i(a^i|\boldsymbol{\tau}_t, \boldsymbol{a}^{-i})$, $\boldsymbol{a}^{-i} := (a^1, \dots, a^{i-1}, a^{i+1}, \dots, a^n)$, for each agent (which is similar to Foerster et al., 2018) with the combined loss $\mathcal{L}_{\text{TD}}$:

$$\mathcal{L}_{\text{TD}} \quad := \quad \mathbb{E}\Big[\tfrac{1}{nT}\sum_{t=0}^T \sum_{i=0}^n \Big(r_t + \gamma q_{\bar{\phi}}^i(\bar{a}_{t+1}^i|\boldsymbol{\tau}_{t+1}, \bar{\boldsymbol{a}}_{t+1}^{-i}) - q_\phi^i(a_t^i|\boldsymbol{\tau}_t, \boldsymbol{a}_t^{-i})\Big)^2\Big] \,, \tag{12}$$

where $\bar{a}_t := \arg\max q_\theta^{\text{VDN}}(\boldsymbol{\tau}_t, \cdot), \forall t$, denotes what a greedy decentralized agent would have chosen. The decentralized value $q_\theta^{\text{VDN}}$ and the greedy difference $v_\psi$, with parameters $\theta$ and $\psi$ respectively, are distilled by regression of the each $q_\phi^i$ in the constraints. First the equality constraint:

$$\mathcal{L}_{\text{OPT}} \quad := \quad \mathbb{E}\Big[ \tfrac{1}{nT}\sum_{t=0}^{T}\sum_{i=0}^{n}\Big(q_\theta^{\text{VDN}}(\boldsymbol{\tau}_t, \bar{a}_t) - \perp q_\phi^i(\bar{a}_t^i|\boldsymbol{\tau}_t, \bar{\boldsymbol{a}}_t^{-i}) + v_\psi(\boldsymbol{\tau}_t)\Big)^2 \Big], \tag{13}$$

where the 'detach' operator $\perp$ stops the gradient flow through $q_\phi^i$. The inequality constraints are more complicated. In principle one would have to compute a loss for every action which has a negative error in (13). Son et al. (2019) suggest to use only the action which minimizes $q^{\text{vdn}}$:

$$\mathcal{L}_{\text{NOPT}} := \mathbb{E}\Big[ \tfrac{1}{nT}\sum_{t=0}^{T}\sum_{i=0}^{n}\Big( \min_{a^i \in \mathcal{A}^i} \Big\{ f^i(a^i|\tau_t^i) + \sum_{j \neq i} f^j(a_t^j|\tau_t^j) - \perp q_\phi^i(a^i|\boldsymbol{\tau}_t, \boldsymbol{a}_t^{-i}) + v_\psi(\boldsymbol{\tau}_t)\Big\}\Big)^2 \Big]. \tag{14}$$

We use this loss, which is called `QTRAN-alt`, as it is reported to perform significantly better. The losses are combined to $\mathcal{L}_{\text{QTRAN}} := \mathcal{L}_{\text{TD}} + \lambda_{\text{OPT}}\mathcal{L}_{\text{OPT}} + \lambda_{\text{NOPT}}\mathcal{L}_{\text{NOPT}}$ , with $\lambda_{\text{OPT}}, \lambda_{\text{NOPT}} > 0$.

**CG** To compare the effect of parameter sharing and restriction to local information in DCG, we evaluate a variation of Castellini et al. (2019) that can solve sequential tasks. In this baseline all agents share a RNN encoder of their belief over the current global state $\boldsymbol{h}_t := h_\psi(\cdot|\boldsymbol{h}_{t-1}, \boldsymbol{o}_t, \boldsymbol{a}_{t-1})$ with $\boldsymbol{h}_0 := h_\psi(\cdot|\boldsymbol{0}, \boldsymbol{o}_0, \boldsymbol{0})$, as introduced in Section 3.1. However, the parameters of the utility or payoff functions are not shared, that is, $\theta := \{\theta_i\}_{i=1}^n$ and $\phi := \{\phi_{ij}|\{i,j\} \in \mathcal{E}\}$. Each set of parameters $\theta_i$ and $\phi_{ij}$ represents one linear layer from $\boldsymbol{h}_t$ to $\mathcal{A}^i$ and $\mathcal{A}^i \times \mathcal{A}^j$ outputs, respectively.

## A.2 HYPER-PARAMETERS

All algorithms are implemented in the PYMARL framework (Samvelyan et al., 2019). We aimed to keep the hyper-parameters close to those given in the framework and consistent for all algorithms.

All tasks used discount factor $\gamma = 0.99$ and $\epsilon$-greedy exploration, which was linearly decayed from $\epsilon = 1$ to $\epsilon = 0.05$ within the first $50,000$ time steps. Every 2000 time steps we evaluated 20 greedy test trajectories with $\epsilon = 0$. Results are plotted by first applying histogram-smoothing (100 bins) to each seed, and then computing the mean and standard error between seeds.

All methods are based on agents' histories, which were individually summarized with $h_\psi$ by conditioning a linear layer of 64 neurons on the current observation and previous action, followed by a ReLU activation and a GRU (Chung et al., 2014) of the same dimensionality. Both layers' parameters are shared amongst agents, which can be identified by a one-hot encoded ID in the input. For the CG baseline, the linear layer and the GRU had $64n = 512$ neurons. This allows a fair comparison with DCG and also had the best final performance amongst tested dimensionalities $\{64, 256, 512, 1024\}$ in the task of Figure 4. Independent value functions $q_\theta^i$ (for IQL), utility functions $f_\theta^v$ (for VDN/QMIX/QTRAN/DCG) and payoff functions $f_\phi^e$ (for DCG) are linear layers from the GRU output to the corresponding number of actions. The hyper-network $\varphi_\phi$ of QMIX produces a mixing network with two layers connected with an ELU activation function, where the weights of each mixing-layer are generated by a linear hyper-layer with 32 neurons conditioned on the global state, that is, the full grid-world. For QTRAN, the critic $q_\phi^i$ computes the $Q$-value for an agent $i$ by taking all agents' GRU outputs, all other agents' one-hot encoded actions, and the one-hot encoded agent ID $i$ as input. The critic contains four successive linear layers with 64 neurons each and ReLU activations between them. The greedy difference $v_\psi$ also conditions on all agents' GRU outputs and uses three successive linear layers with 64 neurons each and ReLU activations between them. We took the loss parameters $\lambda_{\text{OPT}} = \lambda_{\text{NOPT}} = 1$ from (Son et al., 2019) without any hyper-parameter exploration.

All algorithms were trained with one RMSprop gradient step after each observed episode based on a batch of 32 episodes, which always contains the newest, from a replay buffer holding the last 500 episodes. The optimizer uses learning rate 0.0005, $\alpha = 0.99$ and $\epsilon = 0.00001$. Gradients with a norm $\geq 10$ were clipped. The target network parameters were replaced by a copy of the current parameters every 200 episodes.

| Name | Agents | Enemies | Difficulty |
|------|--------|---------|-----------|
| so_many_baneling | 7 Zealots | 32 Banelings | easy |
| 8m_vs_9m | 8 Marines | 9 Marines | hard |
| 3s_vs_5z | 3 Stalker | 5 Zealots | hard |
| 3s5z | 3 Stalker and 5 Zealots | 3 Stalker and 5 Zealots | hard |
| MMM2 | 1 Medivac
2 Marauders
7 Marines | 1 Medivac
3 Marauders
8 Marines | super hard |
| micro_focus | 6 Hydralisks | 8 Stalker | super hard |

Table 2: All tested StarCraft 2 maps for SMAC (Samvelyan et al., 2019).

### A.3 STARCRAFT 2 DETAILS

We kept all hyper-parameters the same and evaluated the six maps in Table 2. All maps are from SMAC (Samvelyan et al., 2019), except micro_focus, which was provided to us by the SMAC authors. The results for DCG-V, DCG, QMIX and VDN are given in Figure 6, where both DCG variants use a rank-1 payoff approximation. Note that our results differ from those in Samvelyan et al. (2019), due to slightly different parameters and an update after every episode. The latter differs from the original publication because we use the the episode_runner instead of the parallel_runner of PYMARL. These choices ended up improving the performance of QMIX significantly.

---

**Algorithm 1** Annotates a CG by computing the utility and payoff tensors (rank $K$ approximation).

---

    **function** ANNOTATE($\{\boldsymbol{h}_{t-1}^i, a_{t-1}^i, o_t^i\}_{i=1}^n, \mathcal{E}, \{\mathcal{A}^i\}_{i=1}^n, K \in \mathbb{N}$)           $\triangleright A := |\cup_i \mathcal{A}^i|$

         $\boldsymbol{f}^{\mathrm{V}} := \boldsymbol{0} \quad \in \quad \mathbb{R}^{n \times A}$              $\triangleright$ initialize utility tensor

         $\boldsymbol{f}^{\mathrm{E}} := \boldsymbol{0} \quad \in \quad \mathbb{R}^{|\mathcal{E}| \times A \times A}$          $\triangleright$ initialize payoff tensor

         **for** $i \in \{1, \ldots, n\}$ **do**              $\triangleright$ compute batch with all agents

             $\boldsymbol{h}_t^i := h_\psi(\boldsymbol{h}_{t-1}^i, o_t^i, a_{t-1}^i)$              $\triangleright$ new hidden state

             $\boldsymbol{f}_i^{\mathrm{V}} \leftarrow f_\theta^v(\boldsymbol{h}_t^i) \quad \in \quad \mathbb{R}^A$              $\triangleright$ compute utility

             **for** $a \in \{1, \ldots, A\} \setminus \mathcal{A}^i$ **do**          $\triangleright$ set unavailable actions ...

                 $\boldsymbol{f}_{ia}^{\mathrm{V}} \leftarrow -\infty$              $\triangleright$ ... to $-\infty$

         **for** $e = (i, j) \in \mathcal{E}$ **do**              $\triangleright$ compute batch with all edges

             **if** $K = 0$ **then**              $\triangleright$ if no low-rank approximation

                 $\boldsymbol{f}_e^{\mathrm{E}} \leftarrow \frac{1}{2} f_\phi^e(\cdot, \cdot | \boldsymbol{h}_t^i, \boldsymbol{h}_t^j) + \frac{1}{2} f_\phi^e(\cdot, \cdot | \boldsymbol{h}_t^j, \boldsymbol{h}_t^i)^\top \quad \in \quad \mathbb{R}^{A \times A}$     $\triangleright$ symmetric payoffs

             **else**              $\triangleright$ if low-rank approximation

                 $[\hat{\mathbf{F}}, \bar{\mathbf{F}}] := f_\phi^e(\cdot, \cdot, \cdot | \boldsymbol{h}_t^i, \boldsymbol{h}_t^j) \quad \in \quad \mathbb{R}^{2 \times A \times K}$

                 $[\hat{\mathbf{F}}', \bar{\mathbf{F}}'] := f_\phi^e(\cdot, \cdot, \cdot | \boldsymbol{h}_t^j, \boldsymbol{h}_t^i) \quad \in \quad \mathbb{R}^{2 \times A \times K}$

                 $\boldsymbol{f}_e^{\mathrm{E}} \leftarrow \frac{1}{2} \hat{\mathbf{F}} \bar{\mathbf{F}}^\top + \frac{1}{2} \bar{\mathbf{F}}' \hat{\mathbf{F}}'^\top \quad \in \quad \mathbb{R}^{A \times A}$          $\triangleright$ symmetric payoffs

         **return** $\{\boldsymbol{h}_t^i\}_{i=1}^n, \boldsymbol{f}^{\mathrm{V}}, \boldsymbol{f}^{\mathrm{E}}$     $\triangleright$ return hidden states $\boldsymbol{h}_t^i$, utility tensor $\boldsymbol{f}^{\mathrm{V}}$ and payoff tensor $\boldsymbol{f}^{\mathrm{E}}$

---

**Algorithm 2** Q-value computed from utility and payoff tensors (and potentially global state $s_t$).

---

    **function** QVALUE($\boldsymbol{f}^{\mathrm{V}} \in \mathbb{R}^{|\mathcal{V}| \times A}, \boldsymbol{f}^{\mathrm{E}} \in \mathbb{R}^{|\mathcal{E}| \times A \times A}, \boldsymbol{a} \in \mathcal{A}, s_t \in \mathcal{S} \cup \{\varnothing\}$)       $\triangleright v_\varphi(\varnothing) = 0$

         **return** $\frac{1}{|\mathcal{V}|} \sum_{i=1}^{|\mathcal{V}|} \boldsymbol{f}_{ia^i}^{\mathrm{V}} + \frac{1}{|\mathcal{E}|} \sum_{e=(i,j) \in \mathcal{E}} \boldsymbol{f}_{ea^i a^j}^{\mathrm{E}} + v_\varphi(s_t)$       $\triangleright$ return the Q-value of the given actions $\boldsymbol{a}$

---

**Algorithm 3** Greedy action selection with $k$ message passes in a coordination graph.

---

    **function** GREEDY($\boldsymbol{f}^{\mathrm{V}} \in \mathbb{R}^{|\mathcal{V}| \times A}, \boldsymbol{f}^{\mathrm{E}} \in \mathbb{R}^{|\mathcal{E}| \times A \times A}, \mathcal{V}, \mathcal{E}, \{\mathcal{A}^i\}_{i=1}^{|\mathcal{V}|}, k$)       $\triangleright A := |\cup_i \mathcal{A}^i|$

         $\boldsymbol{\mu}^0, \bar{\boldsymbol{\mu}}^0 := \boldsymbol{0} \in \mathbb{R}^{|\mathcal{E}| \times A}$          $\triangleright$ messages forward ($\boldsymbol{\mu}$) and backward ($\bar{\boldsymbol{\mu}}$)

         $\boldsymbol{q}^0 := \frac{1}{|\mathcal{V}|} \boldsymbol{f}^{\mathrm{V}}$          $\triangleright$ initialize "Q-value" without messages

         **for** $t \in \{1, \ldots, k\}$ **do**          $\triangleright$ loop with $k$ message passes

             **for** $e = (i, j) \in \mathcal{E}$ **do**          $\triangleright$ update forward and backward messages

                 $\boldsymbol{\mu}_e^t := \max_{a \in \mathcal{A}^i} \left\{ (q_{ia}^{t-1} - \bar{\mu}_{ea}^{t-1}) + \frac{1}{|\mathcal{E}|} \boldsymbol{f}_{ea}^{\mathrm{E}} \right\}$       $\triangleright$ forward: maximize sender

                 $\bar{\boldsymbol{\mu}}_e^t := \max_{a \in \mathcal{A}^j} \left\{ (q_{ja}^{t-1} - \mu_{ea}^{t-1}) + \frac{1}{|\mathcal{E}|} (\boldsymbol{f}_e^{\mathrm{E}\top})_a \right\}$       $\triangleright$ backward: maximizes receiver

                 **if** `message_normalization` **then**       $\triangleright$ to ensure converging messages

                     $\boldsymbol{\mu}_e^t \leftarrow \boldsymbol{\mu}_e^t - \frac{1}{|\mathcal{A}^j|} \sum_{a \in \mathcal{A}^j} \mu_{ea}^t$       $\triangleright$ normalize forward message

                     $\bar{\boldsymbol{\mu}}_e^t \leftarrow \bar{\boldsymbol{\mu}}_e^t - \frac{1}{|\mathcal{A}^i|} \sum_{a \in \mathcal{A}^i} \bar{\mu}_{ea}^t$       $\triangleright$ normalize backward message

             **for** $i \in \mathcal{V}$ **do**          $\triangleright$ update "Q-value" with messages

                 $\boldsymbol{q}_i^t := \frac{1}{|\mathcal{V}|} \boldsymbol{f}_i^{\mathrm{V}} + \sum_{e=(\cdot,i) \in \mathcal{E}} \boldsymbol{\mu}_e^t + \sum_{e=(i,\cdot) \in \mathcal{E}} \bar{\boldsymbol{\mu}}_e^t$       $\triangleright$ utility plus incoming messages

                 $a_i^t := \arg\max_{a \in \mathcal{A}^i} \{q_{ia}^t\}$       $\triangleright$ select greedy action of agent $i$

         **return** $\boldsymbol{a}^k \in \mathcal{A}^1 \times \ldots \times \mathcal{A}^{|\mathcal{V}|}$       $\triangleright$ return actions that maximize the joint Q-value

---

