# OpenReview forum: "Deep Coordination Graphs"
_ICLR.cc/2020/Conference — Reject_

### Official Review · AnonReviewer1 · 2019-10-22
**Official Blind Review #1**

**Rating:** 6

**Review:**

This paper introduces the deep coordination graph for collaborative multi-agent reinforcement learning aimed to solve predator-prey tasks by preventing relative overgeneralization during the exploration of agents.

In general, this paper gives a detailed and comprehensible depiction of the Introduction, Related work and Background section. However, I have two concerns about their method and experiments.

The presentation of the “Method” section is not clear enough to evaluate the contribution of this paper. Specifically:
(1). In the "Method" section, your method incorporates three ideas: 1. restricting the payoffs; 2. sharing parameters; 3. allowing generalization;
In my understanding, both idea 1 and idea 2 come from VDN [4]. Idea 3 is not implemented in this work. Then, what is your contribution?

(2). According to the announcement, the key benefit for your work is "prevent relative overgeneralization during exploration of agents". It is hard to access if the proposed method can prevent overgeneralization. Do you have some theoretical or empirical justifications?

(3). I think it would be helpful if you could give a description according to your algorithm in the appendix. It would be better to draw a diagram to show what is your model.

I think the experiments are too simple and not convincing.
(1). I notice recent multi-agent reinforcement papers ([1], [2] closely related to your work) evaluate their work on the challenging set of StarCraft II micromanagement tasks and achieved the evident result.

(2). In fig.2, why did you only compare your model with VDN [4]?

(3). In fig.3, why does the return value of QTRAN first decrease and then increase? From [3], it seems their return curve continuously grows.

(4). In fig.4, why QTRAN fail to this task?

[1]. Rashid, Tabish, et al. "QMIX: monotonic value function factorisation for deep multi-agent reinforcement learning." arXiv preprint arXiv:1803.11485 (2018).
[2]. Wang, Tonghan, et al. "Learning Nearly Decomposable Value Functions Via Communication Minimization." arXiv preprint arXiv:1910.05366 (2019).
[3]. Son, Kyunghwan, et al. "QTRAN: Learning to Factorize with Transformation for Cooperative Multi-Agent Reinforcement Learning." arXiv preprint arXiv:1905.05408 (2019).
[4]. Sunehag, Peter, et al. "Value-decomposition networks for cooperative multi-agent learning." arXiv preprint arXiv:1706.05296 (2017).

**Experience Assessment:**

I have read many papers in this area.

**Review Assessment: Checking Correctness Of Derivations And Theory:**

N/A

**Review Assessment: Checking Correctness Of Experiments:**

I assessed the sensibility of the experiments.

**Review Assessment: Thoroughness In Paper Reading:**

I read the paper at least twice and used my best judgement in assessing the paper.

---

> ### Author Response · Authors · 2019-11-13
> **Author response**
>
> Thank you for your time and constructive suggestions. We have revised our paper with a greatly extended method and experimental section. We introduce a novel low-rank payoff approximation and a privileged bias function, which allows us to train DCG on StarCraft2 micromanagement tasks. In the following we will address your remarks:
>
> “what is your contribution?”
> As you have mentioned, the ideas for parameter sharing are from [4], however, the application to CG is novel. We added a new baseline (CG, an extension of [Castellini et al.]) in Figures 3 and 4 to demonstrate how much the DCG architecture improves over straight-forward CG.  To strengthen the papers contribution, the uploaded revision also extends the method section significantly.
>
> “It is hard to access if the proposed method can prevent overgeneralization.”
> Relative overgeneralization is the name of a game-theoretic pathology [Panait et al.] and should not be confused with generalization of neural networks. We added a footnote to clarify the difference in the revised version. The task of Figures 3 and 4 is specifically designed to test relative overgeneralization and the success of DCG demonstrates this.
>
> “it would be helpful if you could give a description according to your algorithm in the appendix”
> The revised version adds two algorithms to the appendix: one that computes the utility and payoff tensors and one that computes the value based on these tensors. Together with the algorithm for greedy action selection, this includes all model-specific algorithms.
>
> “evaluate [your] work on the challenging set of StarCraft II micromanagement tasks”
> The new methods of the revised paper were necessary to train DCG on StarCraft2 micromanagement tasks. StarCraft2 results can be found in the (new) Figure 6.
>
> “In fig.2, why did you only compare your model with VDN [4]?”
> In difference to Figure 3, Figure 2 is only supposed to illustrate the dependency of the task on punishment p and the value representation. For p=0 all algorithms learn about equally well (although QTRAN is much slower) and for larger punishment all baseline algorithms break down eventually. We show the latter point in Figure 3, where QMIX, IQL and VDN all receive no reward due to their inability to overcome relative overgeneralization.
>
> “In fig.3, why does the return value of QTRAN first decrease and then increase?”
> [3] shows the same dip (although less pronounced) for the QTRAN-alt curve in Figures 4d and 4e. As the dip is visible for 4 agent, but not for 2, it is reasonable to assume that it becomes more pronounced the more agents the task contains. We evaluate a similar (albeit more complex) task where QTRAN-alt controls 8 agents, and the dip is therefore not inconsistent with the results in [3]. Comparing the time axis in Figure 4d and 4e of [3] with ours also reveals that our implementation recovers rather quickly from the dip.
>
> “In fig.4, why QTRAN fail to this task?”
> QTRAN has theoretical guarantees when optimized with the full set of constraints. This is not possible using neural networks, however, and the approximations QTRAN-base and QTRAN-alt defined in [3] have been proven hard to control in our experiments. While it may be possible to improve the performance slightly with extensive hyper-parameter search, it is unlikely that this would result in a much better performance than e.g. in Figure 3a.

---

### Official Review · AnonReviewer3 · 2019-10-23
**Official Blind Review #3**

**Rating:** 3

**Review:**

1. Summary

Teh authors propose to learn value functions that are a sum of utility (single-agent) and payoff (2-agent) components. The weights between all function are shared (common RNN). This stands in contrast to VDNs (only uses utility functions or centralized value functions. The authors evaluate on predator-prey, where they argue that e.g. VDN fails to learn.

(Note that other work has looked at reward shaping to learn *decentralized* agents for that problem.)

1. Decision (accept or reject) with one or two key reasons for this choice.

Weak reject.

- The comparison between different topologies is nice, but implies that the structure of the graph has to be fixed manually. This seems to be a severe and unscalable constraint. It would be better if the authors would propose and evaluate a method to determine / learn what the right graph structure should be.
- Weight sharing between the various agent components makes the problem closer to a single-agent problem. What happens if the agents are decentralized and the (shared-weight) pairwise functions are separate?
- Authors only evaluate on a predator-prey problem.

4. Supporting arguments

N/A

5. Additional feedback with the aim to improve the paper. Make it clear that these points are here to help, and not necessarily part of your decision assessment.

6. Questions

**Experience Assessment:**

I have published one or two papers in this area.

**Review Assessment: Checking Correctness Of Derivations And Theory:**

N/A

**Review Assessment: Checking Correctness Of Experiments:**

I assessed the sensibility of the experiments.

**Review Assessment: Thoroughness In Paper Reading:**

I read the paper at least twice and used my best judgement in assessing the paper.

---

> ### Author Response · Authors · 2019-11-13
> **Author response**
>
> Thank you for your time and helpful suggestions. We have revised our paper with a greatly extended the method and experimental section. We introduce a novel low-rank payoff approximation and a privileged bias function, which allows us to train DCG on StarCraft2 micromanagement tasks. In the following we will address your remarks:
>
> “that the structure of the graph has to be fixed manually [...] seems to be a severe and unscalable constraint”
> A fully connected graph of n agents has (n^2 - n) / 2 edges. Although this quadratic growth can become infeasible for large n, it is still preferable to the exponential growth of a joint action value function. However, we also provide evidence that a (linearly growing) subset of all edges can yield better performance than pure VDN (e.g. Figure 4b).
>
> “It would be better if the authors would propose and evaluate a method to determine / learn what the right graph structure should be”
> Optimizing graph topologies is usually a NP-hard problem. While the suggestion is welcome, we believe that an efficient algorithm to optimize the topology is beyond the scope of this paper.
>
> “What happens if the agents are decentralized and the (shared-weight) pairwise functions are separate?”
> The paper explicitly does not address tasks that require decentralized execution. We address instead the large class of tasks that can be executed centrally, but is often artificially decentralized to deal with the exponentially growing action space (e.g. StarCraft).
>
> “Authors only evaluate on a predator-prey problem.”
> We extend DCG to work on StarCraft2 micromanagement tasks and show results in the (new) Figure 6.

---

### Official Review · AnonReviewer2 · 2019-10-23
**Official Blind Review #2**

**Rating:** 3

**Review:**

# Summary
This paper proposes a pairwise communication between agents using a shared neural network. The idea is to define the joint action-value function as the sum of individual agent's values + pair-wise payoff between agents, which is based on the prior work [Castellini et al.]. In particular, this paper proposes to share the parameters of the pairwise payoff function to improve efficiency. The result on a grid-world domain shows that the proposed method performs better than baselines that either do not have pairwise communication (VDN) or learn fully joint action value function (QTRAN).

# Originality
The main novelty seems to be coming from the idea of parameter sharing between pairwise payoffs, but the overall architecture seems to be the same as [Castellini et al.].

# Quality
- This paper is missing an important baseline which is DCG without parameter sharing. Given that parameter sharing is the main new component from [Castellini et al.], it would be important to show the benefit of parameter sharing.
- Although the result against several baselines looks good, it would be much more convincing to show some qualitative analysis of the proposed method. For example, showing that the learned payoff captures reasonable and intuitive knowledge would strengthen the paper.

# Clarity
- The paper is well-written, and the figures are very clear.
- It would be better to show a figure that illustrates the domain and task.

# Significance
- Although the paper presents a new idea very well, the overall idea seems a bit incremental. This paper overall looks like a straightforward extension of [Castellini et al.] by adding parameter sharing and evaluating it on a more complex domain. In addition, showing more in-depth analysis would make the paper stronger.

**Experience Assessment:**

I have read many papers in this area.

**Review Assessment: Checking Correctness Of Derivations And Theory:**

I assessed the sensibility of the derivations and theory.

**Review Assessment: Checking Correctness Of Experiments:**

I assessed the sensibility of the experiments.

**Review Assessment: Thoroughness In Paper Reading:**

I read the paper at least twice and used my best judgement in assessing the paper.

---

> ### Author Response · Authors · 2019-11-13
> **Author response**
>
> Thank you for your time and your insightful suggestions. We have uploaded a revised version of the paper with significantly extended method and experimental sections. In the following we will address your remarks:
>
> “This paper is missing an important baseline which is DCG without parameter sharing.”
> Thank you for pointing out this important omission. We added a new baseline (CG) to Figures 3 and 4. The baseline is a straight-forward extension of [Castellini et al.] to sequential tasks: all functions share the same RNN encoder, which conditions on all agents’ histories, but neither utility nor payoff functions share parameters.
>
> “the overall architecture seems to be the same as [Castellini et al.]”
> [Castellini et al.] restricted themselves to non-sequential tasks and did not have to consider how the agents’ histories have to be encoded. The new baseline demonstrates that the input restrictions in DCG have a large effect on performance. Moreover,  we introduce two additional concepts in revised version of the paper: low-rank payoff approximation and a privileged bias function, which are both experimentally verified as well.
>
> “it would be much more convincing to show some qualitative analysis of the proposed method.”
> While the reward of the relative overgeneralization task in Figure 2 and 3 has an interpretable payoff structure, the value that propagates this reward over time has not. One major strength of DCG is that such assumptions do not have to hold in order for the method to work. We would argue that the success in this task is a convincing indirect demonstration that the payoff functions learn what we expect them to.
>
> “the overall idea seems a bit incremental”
> We have greatly extends the scope of the method section and the experimental sections in the revised version.

---

### Decision · Program_Chairs · 2019-12-19

**Decision:**

Reject

**Comment:**

This work extends previous work (Castellini et al) with parameter sharing and  low-rank approximations, for pairwise communication between agents.
However the work as presented here is still considered too incremental, in particular when compared to Castellini et al.
The advances such as parameter sharing and low-rank approximation are good but not enough of a contribution. Authors' efforts to address this concern did not change reviewers' judgment.
Therefore, we recommend rejection.